# New Era of Immune-Based Therapy in Intrahepatic Cholangiocarcinoma

**DOI:** 10.3390/cancers15153993

**Published:** 2023-08-06

**Authors:** Etsushi Kawamura, Tsutomu Matsubara, Norifumi Kawada

**Affiliations:** 1Department of Hepatology, Graduate School of Medicine, Osaka Metropolitan University, Osaka 545-8585, Japan; 2Department of Anatomy and Regenerative Biology, Graduate School of Medicine, Osaka Metropolitan University, Osaka 545-8585, Japan

**Keywords:** intrahepatic cholangiocarcinoma, biliary tract cancer, immunotherapy, immune checkpoint inhibitors, precision medicine

## Abstract

**Simple Summary:**

Intrahepatic cholangiocarcinoma is the second most common primary liver malignancy after hepatocellular carcinoma and accounts for 20% of biliary tract cancers. The incidence of biliary tract cancer has been increasing in recent years, and many patients with biliary tract cancer are diagnosed as unresectable, and, even if resectable, the survival rate is very low. From 2010 to 2021, systemic combination therapy with gemcitabine and cisplatin was the standard therapy for biliary tract cancer. In 2022, the first immunotherapy durvalumab was approved to be added to this combination. As with other solid tumors, immune-based therapies for patients with advanced biliary tract cancer have been shown to have promising outcomes. We discuss the efficacy and safety of these therapies and consider new standards of care.

**Abstract:**

Intrahepatic cholangiocarcinoma (CC) accounts for approximately 20% of all biliary tract cancer (BTC) cases and 10–15% of all primary liver cancer cases. Many patients are diagnosed with unresectable BTC, and, even among patients with resectable BTC, the 5-year survival rate is approximately 20%. The BTC incidence rate is high in Southeast and East Asia and has increased worldwide in recent years. Since 2010, cytotoxic chemotherapy, particularly combination gemcitabine + cisplatin (ABC-02 trial), has been the first-line therapy for patients with BTC. In 2022, a multicenter, double-blind, randomized phase 3 trial (TOPAZ-1 trial) examined the addition of programmed death-ligand 1 immunotherapy (durvalumab) to combination gemcitabine + cisplatin for BTC treatment, resulting in significantly improved survival without notable additional toxicity. As a result of this trial, this three-drug combination has become the new standard first-line therapy, leading to notable advances in BTC management for the first time since 2010. The molecular profiling of BTC has continued to drive the development of new targeted therapies for use when first-line therapies fail. Typically, second-line therapy decisions are based on identified genomic alterations in tumor tissue. Mutations in fibroblast growth factor receptor 1/2/3, isocitrate dehydrogenase 1/2, and neurotrophic tyrosine receptor kinase A/B/C are relatively frequent in intrahepatic CC, and precision medicines are available that can target associated pathways. In this review, we suggest strategies for systemic pharmacotherapy with a focus on intrahepatic CC, in addition to presenting the results and safety outcomes of clinical trials evaluating immune checkpoint inhibitor therapies in BTC.

## 1. Introduction

Biliary tract cancer (BTC) includes intrahepatic cholangiocarcinoma (CC), extrahepatic CC (peri-hilar and distal region), gall bladder cancer, and papillary cancer. BTC affects approximately 210,000 people annually worldwide (approximately 20,000 in Japan), and regional differences in incidence have been reported [1]. Liver flukes are parasites associated with the incidence of BTC, and infection occurs when freshwater fish that serve as hosts are ingested by humans; BTC is considered endemic in the Mekong Basin in Thailand (due to infection by *Opisthorchis viverrini*) and the Xijiang Basin in China (due to infection by *Clonorchis sinensis*) due to dietary habits [2]. The global incidence of BTC, particularly intrahepatic CC, has increased at a rate of 4.4% annually over the last decade [3]. The increase in the incidence of intrahepatic CC has been attributed to both “an increase in the body mass index” and “the inclusion of previously unclassified adenocarcinomas into the ICC category” [4,5]. Intrahepatic CC is a ductal adenocarcinoma that develops in the bile ducts that carry bile from the liver to the duodenum and accounts for 10–15% of all liver cancer cases [6]. Intrahepatic CC is the second-most common type of liver cancer behind hepatocellular carcinoma (HCC). Intrahepatic CC has a poor prognosis, with a 5-year survival rate of 24% after resection [7]. Intrahepatic CC is considered unresectable when patient factors (e.g., hepatic functional reserve or general condition) and tumor factors (e.g., distant metastasis) are poor [8,9].

Since 2010, the universal first-line therapy for unresectable BTC has been systemic chemotherapy using combination gemcitabine + cisplatin (GemCis), with the ABC-02 trial in the UK (*n* = 204) reporting a median overall survival (OS) of 11.7 months and the BT-22 trial in Japan (*n* = 41) reporting a median OS of 11.2 months [10,11]. In 2022, the first immune-based therapy replaced GemCis as the preferred first-line therapy after the TOPAZ-1 trial (*n* = 341) evaluated the addition of durvalumab, an antibody that acts as an immune checkpoint inhibitor (ICI) by targeting programmed death-ligand 1 (PD-L1), to GemCis and reported a median OS of 12.8 months [12]. To guide treatment strategies for intrahepatic CC, various staging systems have been established by the Liver Cancer Study Group of Japan (sixth edition) [13] and the American Joint Committee on Cancer (AJCC, eighth edition) [14,15]. The durvalumab + GemCis trial enrolled patients with unresectable intrahepatic CC with up to Child–Pugh grade A cirrhosis. Another ICI available for BTC therapy is pembrolizumab, an antibody targeting programmed cell death protein-1 (PD-1), which is used as a second-line therapy in cases with high microsatellite instability (MSI). The efficacy of pembrolizumab monotherapy was reported by the KEYNOTE-158 and KEYNOTE-028 trials in 2020.

The overexpression of T cell surface proteins known as immune checkpoints (PD-1, cytotoxic T lymphocyte-associated protein 4 (CTLA-4)), which bind to immune response modulator ligands (PD-L1, B7) on the surface of cancer cells and antigen-presenting cells, promotes the ability of cancer cells to escape the immune system [16]. The blockade of immune checkpoints is an important approach used to activate therapeutic antitumor immunity. PD-L1 expression on the surface of activated macrophages in the hepatic microenvironment leads to a state of chronic immune resistance [17]. Durvalumab and pembrolizumab are ICIs that have demonstrated antitumor efficacy in the hepatic tumor microenvironment and have been approved for the treatment of primary liver cancer. The known therapeutic molecular targets in BTC vary by localization (intrahepatic, extrahepatic, bile ducts, or gall bladders) and have been well summarized by Nakamura et al. (a modified schematic is presented in Figure 1A) [18]. Figure 1B shows the currently available medications that target pathways downstream of the genes (or cell surface receptors) shown in Figure 1A. We previously reported that the Sloan Kettering Institute gene promotes expression of the cyclin-dependent kinase inhibitor tumor protein p21 (TP21) downstream of the TP53 pathway and inhibits the proliferation of intrahepatic human CC cells (Figure 1B) [19]. Precision medicine describes drugs that target specific pathways linked to identified gene mutations, and approximately 40% of BTC cases are expected to have precision medicines based on the results of tumor tissue sampling [20].

Our review focuses on the efficacy and safety of immune-based therapies for the treatment of unresectable BTC cases (especially intrahepatic CC).

## 2. Pathogenesis and Immune Microenvironment of Intrahepatic CC 

The fifth edition of the World Health Organization’s classification of digestive system tumors classifies intrahepatic CCs as either small or large ductal types depending on the cellular origin of the bile duct [21]. Furthermore, Jeon et al. demonstrated that a continuum of histological changes exists between HCC and intrahepatic CC and integrated corresponding features of the immune system and genetic mutations [22]. Small ductal-type CC resembles HCC and is associated with HCC-related etiologies, such as viral hepatitis and metabolic syndrome, whereas large ductal-type CC resembles extrahepatic CC. This is important because the pathologic subclassification of intrahepatic CC is associated with prognostic outcome. We summarize the relationship between the intrahepatic tumor microenvironment and the response to ICI treatment in Figure 2.

The nontumor cells that can be found in the intrahepatic tumor microenvironment originate from immune cells, mesenchymal cells, and vascular cells. The immune system is composed of lymphocytes (T cells, B cells, and natural killer cells) and bone marrow cells (tumor-associated macrophages (TAMs) and bone-marrow-derived suppressor cells (MDSCs)). The principal cells of the mesenchymal lineage are cancer-associated fibroblasts (CAFs). Vascular cells include endothelial cells and pericytes. Crosstalk among signaling pathways activated by cytokines secreted by immunosuppressive cells (TAMs, MDSCs, and regulatory CD4^+^ lymphocytes (Tregs)), antitumor effectors (cytotoxic T cells and natural killer cells), mesenchymal cells, and endothelial cells determines whether tumor cells evade immunosurveillance [23]. Abnormal tumor vasculature prevents the infiltration of CD8^+^ lymphocytes into the tumor core or the delivery of ICI. TAMs are polarized into states that act to promote T cell antitumor potential (TAM1) or promote tumor tolerance (TAM2). In tumors characterized by the presence of immunosuppressive TAM2 cells, vascular endothelial cells are destroyed, forming niches in the vessel wall.

Intrahepatic CC cells can be sensitive (hot) or resistant (cold) to ICI therapy, depending on the tumor microenvironment. Hot tumors respond to ICI due to the presence of CD8^+^ lymphocyte and TAM1 aggregation and low levels of MDSC and CAF infiltration [24]. Cold tumors, by contrast, are characterized by Treg and TAM2 aggregation and high levels of MDSC and CAF infiltration. Cold tumors also present with a more ravaged vascular endothelium than hot tumors, resulting in a worse response to ICI [25]. Jeon et al. theorized that intrahepatic CCs could be categorized into increasingly exhaustive immunotypes, in the order of large ductal type, small ductal type, and HCC-like type, with more exhaustive immunotypes presenting with higher chromosomal instability scores and more likely to be ICI-resistant.

The different ductal types of intrahepatic CC have been associated with different genomic variants. Mutations in *isocitrate dehydrogenase (IDH)1* have been associated with small ductal type, mutations in *Kirsten rat sarcoma (KRAS)* and *smooth muscle actin plus mothers against decapentaplegic 4* have been associated with large ductal type, and mutations in *fibroblast growth factor receptors (FGFR)2* have been associated with intermediate ductal types. Therapies targeting pathways associated with these genes are currently being evaluated in clinical trials, and some have already been applied as precision therapies. Understanding and taking advantage of differences in the origins and tumor microenvironment associated with different forms of intrahepatic CC will improve clinical diagnoses and treatment strategies.

## 3. Immune-Based Therapy for BTC

Treatment of unresectable intrahepatic CC and BTC using ICI therapy alone (targeting PD-1, PD-L1, or CTLA-4) has resulted in low objective response rates (ORRs): 22% for nivolumab alone (targeting PD-1) [26]; 5% for durvalumab alone (targeting PD-L1); 10.8% for durvalumab + tremelimumab (targeting PD-L1 and CTLA-4) [27]; and 3% for nivolumab + ipilimumab (targeting PD-L1 and CTLA-4) [28]. Clinical trials were then designed to evaluate the combination of ICI therapies with cytotoxic anticancer drugs, such as gemcitabine and platinum therapies (Table 1) [29,30]. An overview of the major immunotherapies examined for intrahepatic CC and BTC treatment is provided.

### 3.1. Durvalumab

Durvalumab is a selective human IgG1 monoclonal antibody against PD-L1. In 2020, the US Food and Drug Administration (FDA) approved durvalumab monotherapy for the treatment of urothelial carcinoma, non-small cell lung cancer, and small cell lung cancer. In 2021, the European Medicines Agency approved durvalumab monotherapy for PD-L1–positive non-small cell lung cancer. In 2022, the Pharmaceuticals and Medical Devices Agency of Japan approved durvalumab monotherapy for four cancer types (HCC, BTC, small cell lung cancer, and non-small cell lung cancer).

In 2022, a report was published describing an interim analysis of a multicenter phase 3 trial (TOPAZ-1 trial) evaluating durvalumab + GemCis as first-line therapy for unresectable BTC. Patients with untreated, metastatic, or recurrent disease received durvalumab + GemCis (*n* = 341) or placebo + GemCis (*n* = 344) for up to 24 weeks (8 cycles); starting in the ninth cycle, patients received durvalumab or placebo alone. The OS at 24 months, the primary endpoint, was 24.9% (95% confidence interval (CI): 17.9–32.5%) for the durvalumab arm compared with 10.4% (95% CI: 4.7–18.8%) for the placebo arm (hazard ratio (HR): 0.80 (95% CI: 0.66–0.97), *p* = 0.021)). Secondary endpoints included median progression-free survival (PFS was 7.2 months, HR: 0.75 (95% CI: 0.63–0.89, *p* = 0.001)) and ORR (durvalumab vs. placebo, 26.7% vs. 18.7%). Compared with the placebo arm, the durvalumab arm showed improved OS with a similar safety profile.

In 2022, tremelimumab, a humanized IgG2 monoclonal antibody against CTLA-4, became a standard first-line therapy for unresectable HCC in combination with durvalumab. In the same year, a Korean single-center phase 2 trial (NCT03046862) evaluated the use of quadruple combination therapy, tremelimumab + durvalumab + GemCis (*n* = 32), as second-line therapy in patients with unresectable BTC, which showed acceptable safety; however, the ORR of 70% was inferior to the ORR of 72% reported for durvalumab + GemCis (*n* = 49) [43]. Therefore, tremelimumab is not currently approved for unresectable BTC.

### 3.2. Pembrolizumab

Pembrolizumab is a humanized IgG4 monoclonal antibody against PD-1 that has been approved for the treatment of unresectable malignant melanoma in the US (2014), the UK (2015), and Japan (2016). Pembrolizumab has also been approved for the treatment of PD-L1-positive non-small cell lung cancer in the US (2015) and Japan (2016), in addition to the treatment of Hodgkin lymphoma and urothelial carcinoma.

In 2020, a multicenter phase 2 trial (KEYNOTE-158) conducted in the US, France, and other countries evaluated pembrolizumab monotherapy in patients with MSI-high/deficient mismatch repair (dMMR)-positive non-colorectal solid tumors (*n* = 233). Across the 27 tumor types included in the trial, median PFS was 4.1 months (95% CI: 2.4–4.9 months), ORR was 34.3% (95% CI: 28.3–40.8%), and median OS was 23.5 months (95% CI: less than 13.5 months). Immune-related adverse event (irAE) incidence was 64.8% across all grades, as determined by the Common Terminology Criteria for Adverse Events (CTCAE), whereas the incidence of grades 3–5 irAEs was 14.6% (including one case of grade 5 pneumonia). The KEYNOTE-158 results showed that two of nine BTC patients achieved a partial response [47], and pembrolizumab was approved in 99 countries (2022), including the US (2017) and Japan (2018), for use as second-line therapy in patients with MSI-high BTC refractory or intolerant to prior gemcitabine-based therapies. 

In 2023, a multicenter phase III trial (KEYNOTE-996) conducted in the United States, Japan, Germany, and other countries evaluated pembrolizumab + GemCis (*n* = 533) in patients with untreated, unresectable, locally advanced, or metastatic biliary tract cancer (vs. placebo + GemCis, *n* = 536). There was a statistical improvement in outcomes in the pembrolizumab group compared to the placebo group (median PFS, 6.5 months (95% CI: 5.7–6.9 months) and median OS was 12.7 months (95% CI: 11.5–13.6 months)). This triplet may be a potential new first-line treatment option for patients with unresectable BTC [45]. 

### 3.3. Nivolumab

Nivolumab is a humanized IgG4 monoclonal antibody against PD-1 that was approved in 2014 in Japan and the US for the treatment of advanced malignant melanoma. Subsequently, the drug was approved in Japan, the US, and Europe for the treatment of non-small cell lung cancer (2015) and renal cancer (2016). In 2018, combination nivolumab + ipilimumab (targeting PD-1 and CTLA-4) therapy was approved for malignant melanoma. In 2020, nivolumab + ipilimumab combination therapy was approved in the US for HCC patients previously treated with sorafenib.

In 2019, a Japanese, multicenter, phase 1 trial (JapicCTI-153098; *n* = 30 per arm) evaluated nivolumab + GemCis as first-line therapy for unresectable BTC. Gemcitabine-intolerant patients were converted to nivolumab monotherapy. Gemcitabine-refractory patients were converted to nivolumab + GemCis. In the monotherapy arm, 13% of irAEs were grades 3–5 (grade 5: pleurisy), whereas, in the combination therapy arm, 70% of irAEs were grades 3–5 (grade 5: thrombocytopenia, myocarditis, etc.). The monotherapy arm had a median OS of 5.2 months (95% CI: 4.5–8.7 months), a median PFS of 1.4 months (95% CI: 1.4–1.4 months), and an ORR of 3%; the combination therapy arm had a median OS of 15.4 months (95% CI: 11.8 months to not estimable), a median PFS of 4.2 months (95% CI: 2.8–5.6 months), and an ORR of 37%. This trial showed that nivolumab was well tolerated and safe for use in patients with unresectable BTC [48].

From 2019 to 2022, a group of workers in Japan developed BTC following occupational inhalation of the chlorinated solvent 1,2-dichloropropane, characterized by high PD-L1 expression, which was observed in 100% of these workers (10 of 10 cases) but in less than 10% of individuals with nonoccupational BTC. These workers were treated with nivolumab, and positive results were reported for this case series [49,50]. A Japanese physician-initiated phase 2 trial (OPAL trial) of nivolumab monotherapy in patients with occupational BTC is ongoing (UMIN000034931); however, nivolumab is not currently approved for unresectable BTC due to three reported trials below showing no effect for nivolumab-containing regimens.

In 2020, a US–Korea multicenter phase 2 trial (NCT02829918) evaluated nivolumab monotherapy as a second-line therapy for patients with unresectable BTC. Nivolumab was administered at 240 mg every 2 weeks for a total of 16 weeks, followed by 480 mg every 4 weeks (*n* = 46). ORR was 22%, median PFS was 3.7 months (95% CI: 2.30–5.69 months), and median OS was 14.2 months (95% CI: 5.98 months to not reached). CTCAE grades 3–4 irAEs were reported in 17% of cases (hyponatremia, increased alkaline phosphatase, etc.) [26].

In 2021, an interim analysis of a Taiwanese multicenter phase 2 trial (TCOG T1219 trial; NCT04172402) evaluated nivolumab + gemcitabine + oral 5-fluorouracil (known as capecitabine in the EU and as S-1 in Japan) combination therapy as a first-line therapy for patients with unresectable BTCs. The reported ORR was 41.7%, the median PFS was 9.1 months (95% CI: 7.4 months to not reached), and no data were available for median OS (7.4 months to not achieved).

In 2022, the results of a US multicenter phase 2 trial (BilT- 01 trial, *n* = 33) evaluated combination nivolumab (240 mg every 2 weeks) + ipilimumab (1 mg/kg every 6 weeks) as a second-line therapy for unresectable BTC. The median PFS was 3.9 months, which was comparable to nivolumab monotherapy [28].

### 3.4. Standard Second-Line Therapies after First-Line Therapy Failure

The standard second-line therapies used following the failure of first-line durvalumab + GemCis in Japan and the frequencies at which different genomic targets are identified in intrahepatic CC and BTC are shown in Figure 3 [51,52]. Second-line therapies are selected to target pathways associated with specific gene mutations according to CC characteristics.

For patients who fail durvalumab + GemCis or for whom durvalumab is not indicated, combination 5-fluorouracil + platinum is one therapeutic option. In Europe and the US, 5-fluorouracil + folinic acid + oxaliplatin (FOLFOX) therapy has been approved (following the ABC-06 trial) [40], whereas the JCOG1113 and KHBO1401 trials in Japan revealed that gemcitabine + oral 5-fluorouracil (known as S-1 in Japan) could provide modest survival benefits [33,42]. In 2021, a German phase II trial (NIFE study) reported that nanoliposomal irinotecan combined with 5-fluorouracil/leucovorin is superior to GemCis for the treatment of extrahepatic CC [53]. This combination therapy could potentially be considered a first-line therapy for all forms of BTC in the future.

## 4. Viable Targets and Ongoing Trials for Intrahepatic CC

The reported frequencies of major genetic alterations in intrahepatic CC are 17% for *TP53* (17%), 15% for cyclin-dependent kinase inhibitor 2A/B, 10% for *KRAS*, 20% for *IDH1/2*, 17% for AT-rich interactive domain-containing protein 1A (17%), 15% for *FGFR1/2/3*, and 1% for neurotrophic tyrosine receptor kinase A/B/C (*NTRKA/B/C*) [54]. The primary targets that are currently considered viable options for second-line therapy in unresectable BTC (especially intrahepatic CC) following first-line therapy failure are as follows.

***FGFR1/2/3*** regulates cell differentiation, migration, and proliferation, and mutations in this gene enhance oncogenic signals. Phase 2 trials evaluating the FGFR1/2/3 inhibitors pemigatinib (FIGHT-202 trial) [34], futibatinib (TAS-120 trial) [44], infigratinib [41], delasantinib [55], and erdafitinib (NCT02699606) [56,57] in unresectable BTC characterized by *FGFR1/2/3* fusions or rearrangement reported ORR values of 20–41% and median PFS of 6–7 months. Toxicity profiles include hyperphosphatemia, requiring the use of a low-phosphorus diet with or without phosphochelates; skin disease; and marked mucosal skin dryness. Pemigatinib was approved in Japan, the US, and Europe, whereas futibatinib and infiglatinib were only approved in the US. Phase 3 trials examining the use of pemigatinib (FIGHT-302 trial, NCT036536), futibatinib (FOENIX-CCA3, NCT04093362), and infiglatinib (PROOF trial, NCT03773302) as first-line therapies are underway. Ponatinib (NCT02265341) showed a median PFS of 2.4 months and a median OS of 15.7 months with manageable toxicity in a pilot study [58].

***IDH1/2*** is involved in the generation of NAD+ in the citric acid cycle, regulates normal cell differentiation, and is involved in DNA methylation. Mutations in this gene can inhibit cell differentiation, leading to carcinogenesis. In 2020, phase 3 trials (ClarIDHy; *n* = 124) examined the efficacy and safety of the IDH1 inhibitor ivosidenib, reporting a median PFS of 2.7 months (HR: 0.37; 95% CI: 0.25–0.54, *p* < 0.001) and a median OS of 10.8 months (HR: 0.69, *p* = 0.06). Reported AEs were considered acceptable and primarily characterized as CTCAE grades 1–2 (nausea 33%, diarrhea 31%, fatigue 23%, and cough 21%) [35]. In 2021, ivosidenib was approved for the treatment of unresectable BTC in the US.

***NTRKA/B/C*** regulates neuronal differentiation and maintenance, and mutations in this gene represent significant oncogenic signals. In 2018, a phase 1/2 trial (*n* = 55) examined the efficacy and safety of the NTRK A/B/C inhibitor larotrectinib for the treatment of solid tumors harboring *NRTKA/B/C* fusion mutations. The ORR was 75% for 17 types of cancer, and the partial response rate was 50% among patients with BTC (*n* = 2). The incidence rates for irAEs categorized as CTCAE grades 1–2 (liver injury, dizziness) were 38% in the phase 1 trial and 25% in the phase 2 trial [32]. In the same year, a phase 3 trial (STARTRK-1&2 trial; *n* = 54) examined another NTRK A/B/C inhibitor, entrectinib, reporting an ORR of 57%, and a partial response was reported for the single BTC case (*n* = 1) [37]. Entrectinib was approved in Japan, the US, and Europe, and larotrectinib was approved in the US and Europe.

Other frequently reported mutations in extrahepatic CC have been identified in erythroblastic oncogene B/human epidermal growth factor receptor 2 [31,52], whereas the B-serine/threonine kinase V600E mutation has been reported in BTC [36,51] and the KRAS G12C mutation has been reported in intrahepatic CC [59]. Clinical trials of potential second-line therapies targeting pathways associated with these mutations are currently underway in several countries.

## 5. Predicting ICI Resistance and Countermeasures for BTC

Current biomarkers that are known to predict the response to ICI in patients with solid tumors include PD-L1 expression levels, MSI, MMR, tumor mutation burden, and Wnt/β-catenin signaling [60].

In 2012, preliminary data from patients with advanced solid tumors (melanoma, non-small cell lung cancer, and colorectal cancer; *n* = 236) treated with nivolumab monotherapy demonstrated a relationship between PD-L1 expression in tumor cells and ORR (36%) [61]. As a result, PD-L1 expression in solid tumor biopsies, including primary liver cancer, has been used to predict the efficacy (in terms of improved survival) of anti-PD-1 or anti-PD-L1 therapy [62].

MSI is a genomic mutation phenotype that occurs due to the loss of MMR activity and was first identified in 2015 as a factor with the potential to predict the response to anti-PD-1 therapy, such as pembrolizumab [63]. In 2017, the KEYNOTE-012, 028, 016, 158, 164 trials were conducted across nine countries, including the US, to examine pembrolizumab monotherapy as a second-line therapy in 14 MSI-high/dMMR solid tumors, resulting in a cumulative ORR of 39.6%, including a complete response rate of 7%, leading to FDA approval [64]. In 2020, the phase 2 KEYNOTE-158 clinical trial (which enrolled patients with both PD-L1-negative and -positive tumors) and the phase 1b KEYNOTE-028 clinical trial (which enrolled only patients with PD-L1-positive tumors) showed that the antitumor activity of pembrolizumab did not depend on PD-L1 expression (ORR: 5.8% in KEYNOTE-158; 13% in KEYNOTE-028), and acceptable irAE rates were reported for both trials (CTCAE grades 3–5: 13.5% in KEYNOTE-158, no CTCAE grades 4–5 in KEYNOTE-028) [39].

In 2019, activated Wnt/β-catenin signaling was reported as a promising biomarker for the response to PD-1 therapy in a genetically engineered mouse model of HCC [65]. An ongoing phase 2 clinical trial (201 trial, NCT05091346) is being conducted in four countries, including Japan, to examine combination therapy using the Wnt/β-catenin inhibitor E7386 + pembrolizumab as second-line therapy in patients with HCC. A similar trial may also be conducted for patients with BTC because a 2021 study reported that Wnt/β-catenin inhibitors suppress tumor growth in a BTC mouse model [66]. As shown in Figure 2, cold tumors are resistant to anti-PD-1 therapy (ICI-resistant) and Wnt/β-catenin signaling inhibition in the tumor microenvironment activates antitumor effector T cells, converting cold tumors to hot tumors (ICI-sensitive) [67].

According to an interim analysis of the TOPAZ-1 trial, the tumor area that was PD-L1-positive was smaller in the durvalumab arm than in the placebo arm (60.9% vs. 67.3%), indicating that the prolonged OS observed in the durvalumab arm among patients with BTC was independent of PD-L1 positivity in the tumor tissue. The reported OS values for the trial examining combination durvalumab + tremelimumab + GemCis treatment in unresectable BTC were also similar between patients with and without elevated PD-L1 expression in the tumor (20.0 months vs. 16.0 months, *p* = 0.47) [43]. In the same year, an in vitro study of intrahepatic CC indicated that tumor expression of PD-L1, which is regulated by the interleukin 6-mechanistic target of rapamycin pathway, is associated with poor prognosis [68]. PD-L1 expression is currently a good predictor of immunotherapy response, but the observed heterogeneity of PD-L1 expression within tumors has challenged its reliability [69].

In 2022, the expression of N7-methylguanosine tRNA methyltransferase 1 was identified in the tumor immune microenvironment of intrahepatic CC model mice, where it promotes tumor growth, and the blockade of chemokine pathways downstream of this gene was found to improve the response to PD-1 monotherapy [70].

Current research efforts are focused on investigating additional predictors of the response to ICI in patients with unresectable BTC, in addition to identifying the mechanisms underlying the efficacy of treatments.

## 6. IrAEs in BTC Cases

Since their initial approval, ICIs have become standard therapeutic options for the treatment of unresectable hepatobiliary tumors. However, ICIs are often associated with irAEs that can affect multiple organs. 

### 6.1. Hepatotoxicity

According to the FDA, the incidence of irAE hepatotoxicity can reach 13% (reported for combination ipilimumab + nivolumab therapy) [71]. Among ICIs approved in Japan to treat unresectable BTC, the hepatotoxicity incidence rates for all CTCAE grades are 12% for durvalumab and 0.7% for pembrolizumab, whereas the incidence rates for CTCAE grades 3–4 are 4.8% for durvalumab [72] and 0.14% for pembrolizumab [73]. The potential for hepatotoxicity requires particular attention for patients with hepatobiliary tumors on a background of chronic liver disease.

No established diagnostic criteria based on hepatic histopathology currently exist for identifying hepatotoxic irAE. Therefore, liver biopsies should be performed in cases of irAEs that result in grade 3 or higher liver injury to assess histological severity. Grade 3 or higher liver injury is characterized by serum cytolysis or cholestasis indicators greater than fivefold the upper limit of normal, including alanine aminotransferase > 200 IU/ml and gamma-glutamyltranspeptidase > 300 IU/L in men or 150 IU/L in women. In 2018, a trial (*n* = 563) examining the use of PD-1, PD-L1, or CTLA-4 antibodies to treat solid tumors reported the occurrence of hepatotoxic irAE in 3.5% of cases [74]. All patients with CTCAE grades 3 or higher (*n* = 19, 19/536 = 3.5%) were treated with oral corticosteroids (tapered from 0.5–1 mg/kg/day (7/19 = 37%), maintenance therapy at 0.2 mg/kg/day (2/19 = 11%)), intravenous pulse therapy (methylprednisolone 2.5 mg/kg/day (1/19 = 5%)), or no systemic steroid therapy (6/19 = 32%), and all patients recovered. Systemic steroid therapy may attenuate the antitumor effects of ICIs and should only be administered if the patient fails to improve after ICI therapy discontinuation.

The frequency of hepatotoxic irAE is low, and specific biomarkers are lacking. Therefore, if systemic steroid therapy is not effective, clinicians should consider seeking out rare causes of acute hepatitis (e.g., herpes simplex virus and hepatitis E) and implementing paired liver biopsies (conducted at 2-week intervals). Meticulous attention to detail during the treatment period can lead to the early detection of hepatotoxic irAE and the appropriate introduction of steroid therapy, which can improve the survival of patients receiving immune-based therapy.

### 6.2. Other irAEs

Among AEs related to durvalumab treatment (*n* = 338) in the TOPAZ-1 study [12], 212 cases (63%) were grade 3 or 4 (similar to those in the placebo group (65%)) and 66 cases (20%) were gastrointestinal, including 1 gastrointestinal perforation, 57 liver injuries, 3 cases of pancreatitis, 1 case of colitis, and 4 cases of diarrhea grade 3 or 4. Discontinuation of durvalumab treatment occurred in 30 cases (9%), similar to the placebo group (11%).

## 7. Other Perspectives 

### 7.1. ICI Therapy during the Perioperative Period for BTC

Among patients with resectable BTC, oral anticancer cytotoxic agents, such as 5-fluorouracil-based anticancer agents, used during the postoperative period (BILCAP and ASCOT trials), represent the current standard for adjuvant therapy [75,76]. A current clinical trial in Japan (JCOG 1920 trial) is examining the use of GemCis + 5-fluorouracil as preoperative adjuvant therapy; however, no conclusions have yet been drawn regarding the safety and efficacy of perioperative ICI administration in patients with BTC.

In other cancer types (non-small cell lung cancer, colon cancer with dMMR, and early-stage melanoma), neoadjuvant ICI administration has shown promising survival benefits [77,78,79]. In 2021, the results of a Chinese clinical trial evaluated the PD-1 inhibitor sintilimumab as a postresection, adjuvant, monotherapy option in patients with BTC characterized by dMMR (mutL homolog 1, mutS homolog 2, mutS homolog 6, and postmeiotic segregation increased 2). Although a partial response was detected in 2 of 97 patients (2%) [80], further trials evaluating the preoperative MMR status in patients with BTC are needed.

### 7.2. Steatohepatitis and Primary Liver Cancer

In HCC, the most common form of liver cancer, ICI therapy was found to result in shorter OS among patients with a nonalcoholic fatty liver background compared with other background liver conditions (5.4 vs. 11 months, *p* = 0.023) [81]. Furthermore, an association between the accumulation of CD8^+^PD-1^+^ T cells and HCC progression was also reported in a mouse model of nonalcoholic steatohepatitis [82]. However, no studies have examined the effects of ICI therapy on intrahepatic ICC with respect to the background liver conditions. The increased incidence of fatty liver disease associated with alcohol consumption (risk ratio 3.2) and metabolic syndrome (risk ratio 1.8) is thought to contribute to the worldwide increase in intrahepatic CC incidence [83,84]. Further investigation is warranted.

## 8. Conclusions

The standard therapy for unresectable BTC, including intrahepatic CC, has changed in response to the emergence of immune-based regimens. Although cytotoxic agents, such as GemCis, gemcitabine + 5-fluorouracil, or GemCis + 5-fluorouracil have previously been preferred as first-line therapies, the addition of durvalumab to GemCis in 2022 doubled the 2-year survival rate relative to GemCis alone. Pembrolizumab, an anti-PD-1, is expected to become a first-line agent of next immune-based regimen following durvalumab. Promising clinical trials are currently examining the efficacy of other ICIs for the treatment of unresectable BTC, such as the IMbrave151 trial targeting PD-L1 and vascular endothelial growth factor (atezolizumab + bevacizumab) [46] and the NEOBIL trial examining combination PD-L1 and bintrafusp alfa, which targets *transforming growth factor*-β receptor [38]. Precision medicine has been established as the best option for second-line therapy in intrahepatic CC, in which drugs are selected to target pathways associated with specific mutations, such as those in *FGFR* and *IDH*. The determination of reliable biomarkers remains necessary to identify those patients who will derive the most benefit from and maximize the clinical advantages of immune-based therapies. 

## Figures and Tables

**Figure 1 cancers-15-03993-f001:**
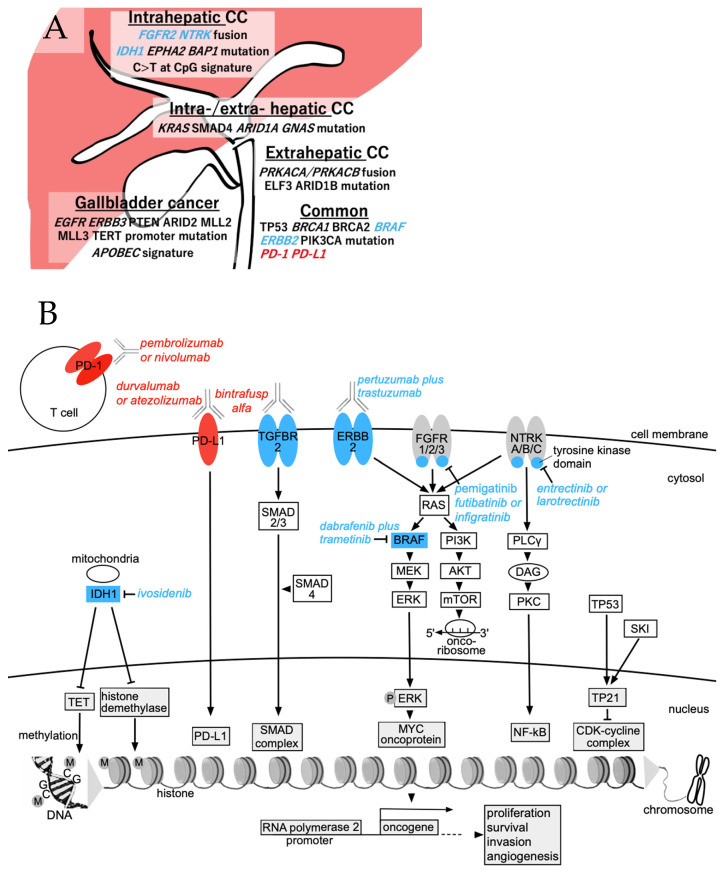
(**A**) The molecular spectrum of biliary tract cancer, adapted from Nakamura et al. [18]. Italics indicate genes associated with a target drug. Extrahepatic cholangiocarcinoma includes peri-hilar and distal cholangiocarcinoma. (**B**) The main therapeutic agents and their targeted pathways in biliary tract cancer, including intrahepatic cholangiocarcinoma. Red, immune checkpoints; blue, genetic alterations; arrows, pathway activation; blocked lines, pathway inhibition. Boxes indicate genes; circles indicate mitochondria, ribosomes, or proteins. Abbreviations: AKT (PKB), protein kinase B; APOBEC, apolipoprotein B mRNA editing enzyme catalytic subunit 3B; ARID, AT-rich interactive domain-containing protein; BAP1, BRCA1 associated protein; BRAF, B-serine/threonine kinase; BRCA, breast cancer susceptibility gene; CC, cholangiocarcinoma; CDK, cyclin-dependent kinase; CpG, cytosine-phosphodiester bond-guanine; DAG, diacylglycerol; EGFR, epidermal growth factor receptor; ELF, E74-like factor; EPHA, erythropoietin-producing hepatocellular receptor A; ERBB, erythroblastic oncogene B; FGFR, fibroblast growth factor receptors; GNAS, guanine nucleotide binding protein, alpha stimulating; IDH, isocitrate dehydrogenase; MEK, mitogen-activated protein kinase/extracellular signal-regulated kinase (ERK) kinase; MLL2 (KMT2D), lysine methyltransferase 2D; MLL3, mixed-lineage leukemia 3; mTOR, mammalian target of rapamycin; MYC, myelocytomatosis oncogene; NFκB, nuclear factor-kappa B; NTRK, neurotrophic tyrosine receptor kinase; P, phosphorylated; PD-L1, programmed death-ligand 1; PI3K, phosphatidylinositol-4,5-bisphosphate 3-kinase catalytic subunit alpha; PKC, protein kinase C; PLC, phosphoinositide-specific phospholipase C; PRKAC, protein kinase A catalytic subunit alpha; PTEN, phosphatase and tensin homolog; RAS (KRAS), Kirsten rat sarcoma; SKI, Sloan Kettering Institute; SMAD, smooth muscle actin plus mothers against decapentaplegic; TERT, telomerase reverse transcriptase; TET, ten eleven translocation; TP53 (21), tumor protein p53 (21).

**Figure 2 cancers-15-03993-f002:**
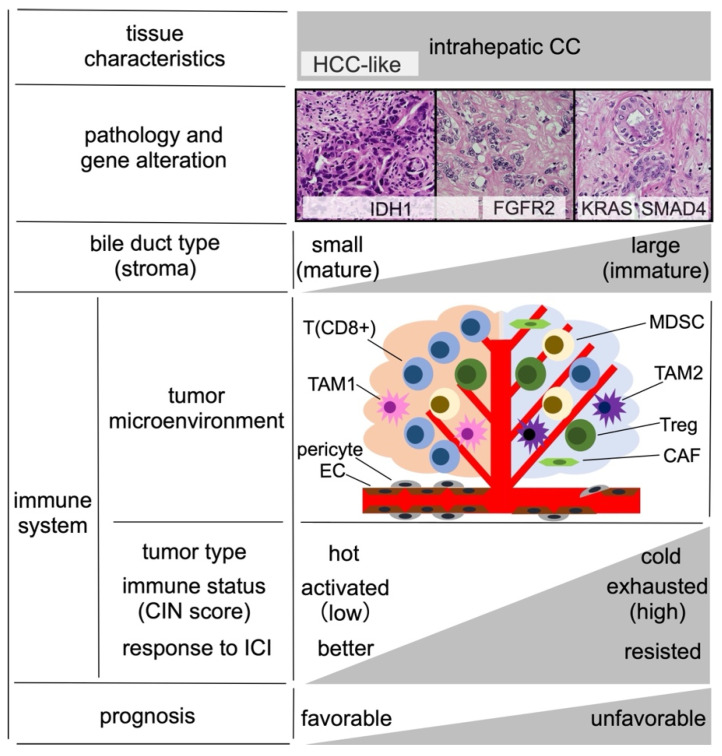
Integration of intrahepatic cholangiocarcinoma pathology and the immune system, adapted from Jeon et al. [22]. Abbreviations: CAF, cancer-associated fibroblast; CC, cholangiocarcinoma; CIN, chromosome instability; EC, endothelial cell; FGFR, fibroblast growth factor receptor; HCC, hepatocellular carcinoma; ICI, immune checkpoint inhibitor; IDH, isocitrate dehydrogenase; KRAS, Kirsten rat sarcoma; SMAD, smooth muscle actin plus mothers against decapentaplegic; MDSC, bone-marrow-derived suppressor cell; PD-L1, programmed death-ligand 1; T (CD8^+^), CD8^+^ T lymphocyte; TAM, tumor-associated macrophage; Treg, regulatory CD4^+^ T lymphocyte.

**Figure 3 cancers-15-03993-f003:**
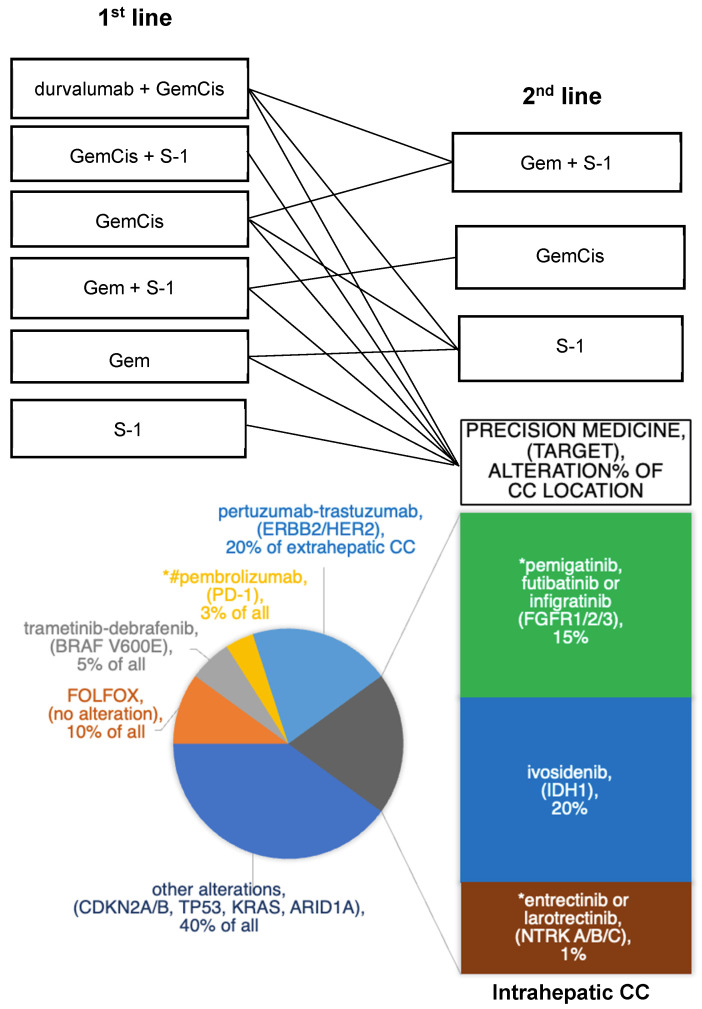
Current standard therapy for unresectable biliary tract cancer, including intrahepatic cholangiocarcinoma, in Japan and the rates of viable genomic targets. The size of the colored sections of this pie chart was determined approximately for each genomic target. The numerical value % within each section is relative to the genetic alteration by location of cholangiocarcinoma onset, including overlaps. Abbreviations: all, intrahepatic/extrahepatic CC and gallbladder cancer; ARID, AT-rich interaction domain; BRAF, B-serine/threonine kinase; CC, cholangiocarcinoma; CDKN, cyclin-dependent kinase inhibitor; Cis, cisplatin; ERBB/HER, erythroblastic oncogene B/human epidermal growth factor receptor; FGFR, fibroblast growth factor receptor; FOLFOX, 5-fluorouracil/folinic acid-oxaliplatin; Gem, gemcitabine; IDH, isocitrate dehydrogenase; KRAS, Kirsten rat sarcoma; NTRK, neurotrophic tyrosine receptor kinase; PD-1, programmed cell death protein 1; S-1, oral 5-fluorouracil (approved only in Japan); #, microsatellite instability-high; *, approved in Japan.

**Table 1 cancers-15-03993-t001:** Major prospective clinical trials with positive results for the treatment of unresectable intrahepatic cholangiocarcinoma and biliary tract cancer.

Regimen	Mechanism	Trial	Phase	Author	Report Year	Patient Number	ORR (%)	Median PFS (Month)	Median OS (Month)	Treatment Line	Conditions	References
GemCis	Cytotoxic anticancer	ABC-02	III	Valle J	2010	204	26	8	11.7	1st		[10]
GemCis	Cytotoxic anticancer	BT22	II	Furuse T	2011	41	20	5.8	11.2	1st		[11]
Pertuzumab + trastuzumab	Anti-ERBB(HER)2	MyPathway	II	Hainsworth JD	2018	7	29	ND	ND	2nd	Gene alteration	[31]
Larotrectinib	Anti-NTRK A/B/C	-	I/II	Drilon A	2018	2	50	ND	ND	2nd	Gene alteration	[32]
GemCis	Cytotoxic anticancer	JCOG1113	III	Morizane C	2019	175	32.4	5.8	13.4	1st/2nd		[33]
Gem + S-1	Cytotoxic anticancer	JCOG1113	III	Morizane C	2019	179	29.8	6.8	15.1	1st/2nd		[33]
Pemigatinib	Anti-FGFR1/2/3	FIGHT-202	II	Abou-Alfa GK	2020	107	35.5	6.9	21.1	2nd	Gene alteration	[34]
Ivosidenib	Anti-IDH1	ClarIDHy	III	Abou-Alfa GK	2020	124	2.4	2.7	10.8	2nd	Gene alteration	[35]
Dabrafenib + trametinib	Anti-B-Raf + anti-MEK	ROAR	II	Subbiah V	2020	43	47	9	14	2nd	Gene alteration	[36]
Entrectinib	Anti-NTRK A/B/C	STARTRK-1&2	I/II	Doebele RC	2020	1	100	ND	ND	2nd	Gene alteration	[37]
Bintrafusp alfa	Anti-PD-L1 + anti-TGFBR2	NEOBIL	I	Yoo C	2020	30	20	2.5	12.7	2nd		[38]
Pembrolizumab	Anti-PD-1	KEYNOTE-028	Ib	Piha-Paul SA	2020	24	13	1.8	5.7	2nd	MSI-high, PD-L1-positive	[39]
Pembrolizumab	Anti-PD-1	KEYNOTE-158	II	Piha-Paul SA	2020	104	5.8	2	7.4	2nd	MSI-high	[39]
Folinic acid + fluorouracil + oxaliplatin	Cytotoxic anticancer	ABC-06	III	Lamarca A	2021	81	5	ND	6.2	2nd		[40]
Infigratinib	Anti-FGFR1/2/3	-	II	Javle M	2021	122	23.1	6.7	ND	2nd	Gene alteration	[41]
Nivolumab	Anti-PD-1	OPAL	II	ND	2022	16	ND	ND	ND	2nd	* Interim analysis	ND
GemCis + S-1	Cytotoxic anticancer	KHBO1401	III	Ioka T	2022	123	41.5	7.4	13.5	1st		[42]
Durvalumab + GemCis	Cytotoxic anticancer + anti-PD-L1	-	II	Oh DY	2022	49	72	11	18.1	1st		[43]
Durvalumab + GemCis	Cytotoxic anticancer + anti-PD-L1	TOPAZ-1	III	Oh DY	2022	341	26.7	7.2	12.8	1st	Interim analysis	[12]
Futibatinib	Anti-FGFR1/2/3	TAS-120	II	Goyal L	2023	103	42	9	21.7	2nd	Gene alteration	[44]
Pembrolizumab + GemCis	Anti-PD-1 + cytotoxic anticancer	KEYNOTE-966	III	Kelley RK	2023	533	ND	6.5	12.7	1st		[45]
Atezolizumab + bevacizumab + GemCis	Cytotoxic anticancer + anti-PD-L1 + anti-VEGF	IMbrave151	II	Shemesh CS	2023	78	ND	ND	ND	1st	Interim analysis	[46]

Blue, precision medicine; red, immune-checkpoint inhibitor; Cis, cisplatin; CTLA, cytotoxic T lymphocyte-associated protein; ERBB, erythroblastic oncogene B; FGFR, fibroblast growth factor receptor; Gem, gemcitabine; HER, human epidermal growth factor receptor; IDH, isocitrate dehydrogenase; MEK, mitogen-activated protein kinase/extracellular signal-regulated kinase kinase; MSI, microsatellite instability; ND, no data; NTRK, neurotrophic tyrosine receptor kinase; ORR, objective response rate; OS, overall survival; PD-1, programmed cell death protein 1; PD-L1, programmed death-ligand 1; PFS, progression-free survival; Raf, serine/threonine-protein kinase; S-1, oral 5-fluorouracil; TGFBR, transforming growth factor beta receptor; * occurred at a printing company (chlorinated solvent) in Osaka, Japan (UMIN000034931).

## Data Availability

The data can be shared up on request.

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
