# Peer review of "New Era of Immune-Based Therapy in Intrahepatic Cholangiocarcinoma"

_cancers, 2023, doi:10.3390/cancers15153993_

Round 1

Reviewer 1 Report

Kawamura et al. reviewed the immune-based systemic drug therapy for intrahepatic CC, which appears to be timely as the GemCis plus ICI therapy is currently available in the clinical setting for BTC. This review article covered a wide range of topics concerning anticancer therapy for intrahepatic CC, such as molecular profiling, therapeutic agents and their targeted signaling pathways, immune microenvironment, immune classification, as well as the results of latest clinical trials and molecular-targeted therapy based on the genetic alterations. The content of this review article would be substantial and satisfactory. However, there are a few comments to be addressed as follows.

Specific comments;

 1. The systemic drug therapy has been carried out in the entity of BTC including intrahepatic CC, extrahepatic CC and gall bladder cancer. The latest GemCis plus ICI therapy is approved for patients with BTC. There has been accumulating evidence as to the genomic alterations of BTC, resulting in clarification of significant differences among intrahepatic CC, extrahepatic CC and gall bladder cancer. In response to this, anticancer therapy for BTC has been considered separately by cancer localization; especially intrahepatic CC is becoming an independent disease. Indeed, FGFR inhibitors have been regarded as agents for intrahepatic CC but not for other BTC.

   In this regard, the authors should distinguish more clearly between the contents for BTC and those for intrahepatic CC only throughout the article.

2. lines 325; is the positive rate for the genomic alteration of NTRK so high in intrahepatic CC?

3. 7.2 Background liver characteristics and the effect of ICI; the content would be applicable for HCC but not for intrahepatic CC.

Reviewer 2 Report

The authors in this manuscript discuss a very important and recent topic which is the therapeutic role of immune therapy in unresectable intrahepatic cholangiocarcinoma. The manuscript is well written and organized. The figures are representative and easy to be understood. Minor comments are to be addressed:

1. Mention how the BTC is considered unresectable.

2. In line 132, change the title from 'pathology' into 'pathogenesis' because the next paragraphs focus more on the mechanism rather than pathology only. 

Reviewer 3 Report

The authors have created a review on the role of immunotherapy on intrahepatic cholangiocarcinomas, a sub-category of biliary tract cancers. The authors have summarized much of the literature with some focus on Asia where the prevelence is highest. 

After reading this, some questions and comments did arise including:

1.  In the introduction, BTC and intrahepatics seem to be used interchangeably. It might be good to introduce that BTC comprises of extrahepatics and gall bladder. 

2. In figure 1, where are the perihilar tumors on the figure?

3. On page 4, line 133-135, there is mention of small or large ductal types but the importance of this is not mentioned. I would add a line to explain why this is important, otherwise why bring it up?

4. In Keynote 966, you mentioned the AE death from pembro but not from the study arm. for proper comparison, you should mention both.

5. There is a fair amount of data for FGFR2 targeted therapies. It feels glossed over in this review and I would want to have it expanded further. 

6. There are some citations I would have expected in a manuscript like this including: 

        a. Rizzo A, Ricci AD, Brandi G. Recent advances of immunotherapy for biliary tract cancer. Expert Rev Gastroenterol Hepatol. 2021 May;15(5):527-536. doi: 10.1080/17474124.2021.1853527. Epub 2021 Jan 8. PMID: 33215952.

        b. Scott AJ, Sharman R, Shroff RT. Precision Medicine in Biliary Tract Cancer. J Clin Oncol. 2022 Aug 20;40(24):2716-2734. doi: 10.1200/JCO.21.02576. Epub 2022 Jul 15. PMID: 35839428.

        c. Woods E, Le D, Jakka BK, Manne A. Changing Landscape of Systemic Therapy in Biliary Tract Cancer. Cancers (Basel). 2022 Apr 25;14(9):2137. doi: 10.3390/cancers14092137. PMID: 35565266; PMCID: PMC9105885.

        d. Tam VC, Ramjeesingh R, Burkes R, Yoshida EM, Doucette S, Lim HJ. Emerging Systemic Therapies in Advanced Unresectable Biliary Tract Cancer: Review and Canadian Perspective. Curr Oncol. 2022 Sep 28;29(10):7072-7085. doi: 10.3390/curroncol29100555. PMID: 36290832; PMCID: PMC9600578.

I feel a more exhaustive literature search would be required to beef up this manuscript.

7. Figure 1A, the blue font of FGFR2, NRTK overlaid on the pink liver is difficult to view. Might suggest a different color.

8. Table 1 is completely unreadable. It needs to be enlarged a fair bit in order to be read.

9 As many BTC treaters have not used immunotherapy much (as it is only just entering in the GI tumor space), this would be a good opportunity to discuss IO toxicities that might be specific or seen more common in GI (beyond the hepatitis).

From a writing perspective, there were a couple of things that I noted.

1. On page 2, line 88, there is a font change that should be corrected.

2. On page 6, starting line 238 there is another font change that needs to be corrected. 

Reviewer 4 Report

Title: New era of immune-based therapy in intrahepatic cholangiocarcinoma                               

This paper describes strategies for systematic pharmacotherapy with a focus on intrahepatic CC as well as the results and safety outcomes of clinical trials evaluating immune checkpoint inhibitor therapies in BTC.

The manuscript cites many references and is well written. Author`s responses to the reviewer's comments were thorough and well-written. 

There is a one question the authors should answer.

1.     Durvalumab

Durvalumab has several side effects.

The authors should describe these details.

Reviewer 5 Report

The irAEs except hepatic toxcity should also be discussed in detail.
